# Tuning Electromagnetic Parameters Induced by Synergistic Dual-Polarization Enhancement Mechanisms with Amorphous Cobalt Phosphide with Phosphorus Vacancies for Excellent Electromagnetic Wave Dissipation Performance

**DOI:** 10.3390/nano13233025

**Published:** 2023-11-27

**Authors:** Bo Wen, Yunzi Miao, Zhijie Zhang, Na Li, Jiyuan Xiao, Yushuo Li, Jiangtao Feng, Shujiang Ding, Guorui Yang

**Affiliations:** 1School of Chemistry, Xi’an Jiaotong University, Xi’an 710049, China; wenbom@stu.xjtu.edu.cn (B.W.); myznofwplz@163.com (Y.M.); lina0831@xjtu.edu.cn (N.L.); xjy@nwafu.edu.cn (J.X.); lys11544@163.com (Y.L.); dingsj@mail.xjtu.edu.cn (S.D.); 2Engineering Research Center of Energy Storage Materials and Devices, Ministry of Education, Four Joint Subjects One Union, Xi’an Jiaotong University, Xi’an 710049, China; 3School-Enterprise Joint Research Center for Power Battery Recycling & Circulation Utilization Technology, Xi’an Jiaotong University, Xi’an 710049, China; 4Department of Environmental Science and Engineering, Xi’an Jiaotong University, Xi’an 710049, China; zzhijie@stu.xjtu.edu.cn (Z.Z.);

**Keywords:** Co metal–organic framework, amorphous CoP_x_/carbon composites, phosphorus vacancy, dipole polarization, defect polarization, electromagnetic wave absorption

## Abstract

The understanding of amorphous and heterojunction materials has been widely used in the field of electromagnetic wave absorption due to their unique physical and chemical properties. However, the effectiveness of individual strategies currently used is still limited. Well-designed compositions and amorphous structures simplify the effect of different polarization mechanisms on the absorption of electromagnetic waves. In this work, through the carbonization and controlled phosphating of one-dimensional Co Metal–Organic Framework (Co-MOF) nanorods, the synthesis of complex components and amorphous CoP_x_ with phosphorus vacancies is successfully accomplished, thus adjusting the optimization of electromagnetic parameters. Phosphorus-vacancy-induced defective polarization loss and enhanced-electronegativity-differences-induced dipole polarization loss synergistically as a dual-polarization strategy significantly improved the electromagnetic parameters and impedance matching. In conclusion, the electromagnetic parameters of the Co@CoP_x_@C composites are indeed significantly regulated, with reflection losses of −55 dB and a bandwidth of up to 5.5 GHz. These innovative research ideas provide instructive thinking for the development of amorphous absorbers with vacancies.

## 1. Introduction

Electromagnetic (EM) wave absorbents possess great application potential in preventing EM interference and human health damage due to a reliable EM wave attenuation ability, resulting in the emergence of a large number of new materials and revealing avant-garde EM wave attenuation mechanisms in recent years [1,2,3,4,5]. Commonly, the attenuation mechanisms include conduction loss, magnetic loss, and dielectric loss, as well as the multiple reflection and scattering of EM waves caused by microstructures [6,7,8]. Dielectric losses generally rely on various polarization losses to provide strong attenuation EM wave characteristics, while magnetic losses rely on resonance and eddy current to provide smaller levels of EM wave attenuation capabilities [9,10]. The magnetic loss capacity is generally much lower than the dielectric loss, and the significance of it existing is that it can be used to improve impedance matching and promote EM wave absorption [11]. Conduction loss, multiple reflections, and scattering generally only serve to promote the EM wave absorption [12]. Therefore, the hot research in the field of EM wave absorption focuses on the regulation of multi-component and complex structures, leading to the facilitation of the attenuation characteristics of EM waves by using complex and changeable loss mechanisms [13,14].

To enrich the types of loss mechanisms in absorbents, the composition and structure regulation of absorbents are commonly used strategies. Naturally, absorbents with superior electrical conductivity, including graphene, carbon nanotubes, and conductive polymers, can rely on excellent electrical conductivity and functional groups to contribute to favorable conductive loss and polarization loss [15,16,17]. Magnetic materials include magnetic metal elements, alloys, and ferrite materials, in which magnetic metal elements and alloys generally only provide magnetic loss, while ferrite materials also provide additional dielectric loss [18,19,20]. A large number of studies have proved that in magnetic materials/graphene, magnetic materials/carbon nanotubes, and magnetic materials/conductive polymers, the abundant heterojunctions promote enhanced interfacial polarization loss [21]. Of course, in addition to the above composite materials, oxides/carbon sulfide/carbon, selenide/carbon, and other composite materials have also become a class of materials that contribute to EM wave attenuation [22,23,24]. Therefore, combined with the above analysis, the combination of different types of materials can achieve the enhancement of EM wave absorption ability. However, the above-mentioned magnetic composite material has poor corrosion resistance and a single heterogeneous structure, which leads to a large number of limitations in the actual use process.

At present, due to their high intrinsic magnetism, relatively higher electrical conductivity, corrosion resistance, and preferable stability, the transition metal phosphide heterojunctions are widely used in catalysis, batteries, sensors, and other fields, but there is still less research in the field of EM wave absorption [25,26,27]. The excellent properties of heterojunctions are attributed to the following explanations. First, charge transfer at the complex heterojunction interfaces results in a reallocation of electrons, which improves the interfacial polarization ability. Second, the complex heterogeneous structure can realize the regulation of the defect structure and help to improve the polarization of the defects. Third, the abundant edge activity is also conducive to electron transfer, which enhances the interfacial polarization loss. For example, Fan et al. [28] prepared CoP/RGO composites with a reflection loss (RL) value of −52.60 dB at 2.05 mm through the synergistic enhancement of conductive loss and polarization loss. Furthermore, constructing a synergistic combination of multiple heterojunctions is another strategy to improve the attenuation of EM waves, as they tend to have better synergistic attenuation. Wang et al. [29] synthesized 3D flower-like CoNi-P/C composites that exhibited an optimal RL value of −65.5 dB depending on the excellent dielectric loss such as conductive loss and polarization loss. However, most of the above heterojunctions are crystal–crystalline heterojunctions, and crystal–amorphous heterojunctions are less studied. Moreover, the further formation of an amorphous structure is conducive to further increasing the defect loss and internal electron transfer.

Herein, we successfully synthesized amorphous Co@CoP_x_@C composites with phosphorus vacancies with the controlled phosphating of MOF-derived metal/carbon composites. After partial phosphating, the impact of cobalt metal radioactivity on living organisms and the impact on crop growth can be reduced. Depending on the charge transfer at the interface of the multiple heterojunctions, the interfacial polarization loss is enhanced. The constructed phosphorus vacancies also promote the formation of more defect polarization. Additionally, phosphorus-doped porous carbon is also proved to be an important cause of enhanced polarization loss. Benefitting from the above improvement in polarization loss, the prepared Co@CoP_x_@C composites realize the effective attenuation of EM waves with lower RL characteristics (−55 dB) and a wider absorption bandwidth (5.5 GHz). This work on the construction of amorphous heterojunctions with abundant phosphorus vacancies reveals the mechanism of polarization and defect effects on EM wave absorption characteristics and provides a new strategy for further designing EM wave absorbents with excellent loss capacity by depending on their enhanced polarization and defect loss capabilities.

## 2. Experimental Section

### 2.1. Materials

Cobalt acetate tetrahydrate (Co(Ac)_2_·4H_2_O, AR, 98.5%), benzene-1, 3, 5-tricarboxylic acid (H_3_BTC, 98%), and Selenium sulfide (SeS_2_, 99%) were purchased from Sinopharm Chemical Reagent Co., Ltd., Shanghai, China.

### 2.2. Preparation of Co-MOF Nanorods

The Co-MOF nanorods were synthesized with the following methods. First, 4 mmol Co(Ac)_2_·4H_2_O was dissolved in 40 mL deionized water to form a clarified solution. Then, 4 mmol H_3_BTC was added into the above solution, and the solution was stirred for 12 h. Finally, the product was washed with ethanol and deionized water before being dried at 60 °C overnight. 

### 2.3. Preparation of Co@C and Co@CoP_x_@C Composites

The synthesized Co-MOF nanorods were put in a quartz boat and annealed at 700 °C for 2 h with a heat rate of 2.0 °C min^−1^ under an argon atmosphere; then, Co@C composite material was obtained. Further, the Co@CoP_x_@C composites were obtained by placing different molar quantities of NaH_2_PO_2_·H_2_O (1, 5, 10) and Co@C at the left and right ends of a porcelain boat and calcining at 300 °C under the protection of argon for 2 h.

### 2.4. Characterization

The crystal structures were measured using X-ray diffraction (XRD, D8 Advance, Germany). A field-emission scanning electron microscope (FE-SEM, GeminiSEM 500, Oberkochen, Germany) and transmission electron microscope (TEM, ThermoFisher, Talos-F200X JEM-F200, Waltham, MA, USA) were used to confirm the morphology of the composites. The surface composition of composites was analyzed with X-ray photoelectron spectra (XPS, KRATOS, AXIS ULtrabld). The Raman (Raman, Renishaw, inVia Qontor) characterization test demonstrates the graphitization degree of the products. The specific surface area was obtained based on the Brunauer–Emmett–Teller (BET) theory (ASAP 2020 device). 

## 3. Results and Discussion

Figure 1a demonstrates the schematic diagram of the generation of Co@CoP_x_@C composite material after a multi-step reaction. Co-MOF nanorods were prepared with the stirring method at room temperature, as described in the experimental section. Co@CoP_x_@C composites can be obtained by first carbonizing the Co-MOF nanorods and then placing NaH_2_PO_2_·H_2_O and Co@C in the upstream and downstream of the porcelain boat with different contents of NaH_2_PO_2_·H_2_O for low-temperature calcination, respectively.

To prove that Co-MOF nanorods were successfully synthesized, the X-ray diffraction (XRD) pattern is depicted in Figure 1b, demonstrating that the prepared precursors were Co-MOF nanorods with good crystallinity. Co-MOF nanorods with smooth surfaces were detected with SEM images, as shown in Figure 1c. The SEM images in Figure 1d verify that the Co@C composite can maintain the original nanorods’ morphology after calcination with an apparently coarse surface. On top of that, the phosphating samples remained rod-like with rougher surfaces (Figure 1e–g). 

Further analysis of the crystal structure of Co@C composites using XRD patterns testified to the presence of three diffraction peaks at 44.4°, 51.7°, and 76.0°, corresponding to the (111), (200), and (220) crystal planes of Co, respectively (JCPDS No. 15-0806) (Figure 2a) [30]. The intensity of the diffraction peaks on the (111) crystal plane of Co decreased with the increase in the degree of phosphatization, and the two crystal planes (200) and (220) almost disappeared. Herein, attributed to the possibility of the existence of amorphous CoP_x_, the diffraction peaks related to CoP_x_ were not clearly found on the diffraction peaks of the phosphated samples. P doping is an effective method for introducing defects in carbon substrates. As displayed in Figure 2b, I_D_/I_G_ values reveal that two peaks located at about 1340 and 1590 cm^−1^ are associated with the D-band and G-band of the carbon, respectively [31,32]. The relative intensity values of I_D_/I_G_ in different Co@ CoP_x_ @C composites are slightly higher than those of Co@C composites, facilitating the provision of reliable evidence of increased defects in the carbon. 

Corroborating the existence of phosphorus vacancies in Co@CoP_x_@C composites, electron paramagnetic resonance (EPR) was reliably used as a direct and effective technique for recognizing phosphorus vacancies [33]. As detected in the analysis results, the presence of phosphorous vacancies implies that the vacancy-induced defect polarization loss will appear (Figure 2c). Transmission electron microscopy (TEM) images of Co@CoP_x_@C composites ascertain that with the increasing degree of phosphatization, the appearance of visible large pore structures on the surface of the nanorods may be caused by the loss of carbon during the phosphatization process (Figure 2d–f). The HRTEM image flags up that the interplanar distance of 0.205 corresponds to the Co nanoparticles (111) plane, proving the presence of Co particles in Co@CoP_x_@C-5 composites (Figure 2g) [34]. The high-angle annular dark-field (HAADF) images and the corresponding energy spectrometer (EDS) mapping images confirm the presence of the elements C, P, and Co (Figure 2h–k). In contrast, the particle sizes of different samples can be statistically derived from Figure 2d–f and Appendix A. The average particle sizes of 5.13, 10.45, 11.77, and 11.9 nm in Co@C, Co@CoP_x_@C-1, Co@CoP_x_@C-5, and Co@CoP_x_@C-10 composites testify to the markedly increased particle sizes of the phosphated samples (Appendix A). The pore characteristics of Co@C and Co@CoP_x_@C composites were analyzed using nitrogen adsorption–desorption isotherms. 

Specific surface area measurement is necessary to determine the structure of the samples after phosphating as both the particle size and carbon content of the samples are significantly altered. The adsorption–desorption curves in Appendix A show that the Co@C, Co@CoP_x_@C-1, Co@CoP_x_@C-5, and Co@CoP_x_@C-10 composites can be found to have the same adsorption–desorption curves, all of which are of type IV isotherms showing numerous mesoporous structures. Based on the calculations, the specific surface areas of Co@C, Co@CoP_x_@C-1, Co@CoP_x_@C-5, and Co@CoP_x_@C-10 composites are 71.856, 37.693, 54.933, and 131.632 m^2^/g, with the average pore size distributions of 1.90, 2.02, 2.32, and 6.06 nm (calculated with the BJH method) (Appendix A). It is deduced that the specific surface areas of Co@CoP_x_@C-1 and Co@CoP_x_@C-5 composites are significantly lower than that of Co@C composites due to the increase in particle size after partial phosphatization. The specific surface area of Co@CoP_x_@C-10 composites is larger than that of Co@C composites, which comes from the fact that a greater degree of phosphatization can result in the volatilization of the carbon component.

The transmutation of the element valence and composition for the products were revealed by X-ray photoelectron spectroscopy (XPS) [35]. As shown in Figure 3(a1–d1), the three distinctive elements C, Co, and accessional P can be detected in the survey spectrum of the samples. The comparable C1s of Co@C, Co@CoP_x_@C-1, Co@CoP_x_@C-5, and Co@CoP_x_@C-10 composites speculate the inability of the phosphorylation process to significantly modify the structure of the carbon, and and C-C/C=C C-O, C=O (284.8, 285.6, 289.0 eV) peaks exist in the carbon structure for all samples exist in the carbon structure for all samples (Figure 3(a2–d2)) [36]. The typical XPS peaks of the metallic Co fraction also decreased significantly with the intensification of the phosphatization process and are divided into several peaks (Co^0^ 778.0 eV and 794.2 eV; Co^2+^ 781.7 eV and 797.5 eV; satellite peaks 802.5) (Figure 3(a3–d3)), indicating the gradual conversion of metallic Co to CoP [37]. As for the spectrum of P 2p, the peak at a binding energy of 130.2 eV stems from the 2P electron of phosphorus in CoP. For Co@CoP_x_@C-1 composites, the absence of P-O is attributed to the fact that the less phosphated sample is protected by a carbon layer (Appendix A) [38]. As the Co@CoP_x_@C-5 and Co@CoP_x_@C-10 composites are exposed to air, an oxidation reaction occurs on the surface, producing another peak attributed to P-O at 134.1 eV (Appendix A). 

Commonly, the complex permittivity (ε_r_ = ε′ − jε″) and permeability (μ_r_ = μ′ − jμ″) are closely correlated with the mechanism affecting the attenuation of EM waves [39,40,41]. Analysis of the differences in the EM parameters of the samples is authoritative in supervising the excogitation of absorbents with preeminent electromagnetic wave absorption properties. Considering the presence of dispersion behavior, the incremental frequency will induce a decrease in the ε′ values [42]. Consequently, the curve trend analysis based on Figure 4a deduces that the ε′ values of all samples are expected to drop from 9.57, 14.95, 12.73, and 9.95 to 4.22, 5.83, 6.22, and 4.78 with increasing frequency, where the ε′ values of Co@CoP_x_@C-1 and Co@CoP_x_@C-5 composites are relatively larger, connoting an eminent charge storage capacity. The cause for the decrease in the ε′ values of the Co@CoP_x_@C-1, Co@CoP_x_@C-5, and Co@CoP_x_@C-10 composites can be attributed to the decrease in carbon content and the increase in specific surface area. According to the free electron theory, the variation in ε″ value is closely related to the electrical conductivity. As presented in Figure 4b, the difference in the presented curves infers that the ε″ value gradually decreases with increasing phosphorylation, which may be due to the decrease in conductivity. The conductivity test can also prove that the declining conductivity of the phosphated products is accompanied by the growth of the phosphating degree [43]. Typically, the Debye polarization relaxation model is relied upon to identify polarization loss and conduction loss. The ε′-ε″ can be expressed as the following equation [44,45,46]:(1)$${\left(\varepsilon^{’}-\frac{{\varepsilon_{s}+\varepsilon }_{\infty }}{2}\right)}^{2}+{\left(\varepsilon \prime \prime \right)}^{2}={\left(\frac{\varepsilon_{s}-\varepsilon_{\infty }}{2}\right)}^{2}$$

The Cole–Cole curve obtained by analyzing the relationship between ε′ and ε′ is plotted in Appendix A, where each Debye relaxation process comes from a semicircular. It is deduced that the presence of multiple semicircular portions in the low-frequency region and the apparent presence of straight portions in the high-frequency region in Co@C and Co@CoP_x_@C-1 composites prove the presence of conduction loss and multiple polarization loss mechanisms. The presence of only multiple semicirculars in Co@CoP_x_@C-5 and Co@CoP_x_@C-10 composites suggests the predominance of polarization loss.

To further visualize the attenuation degree of conductivity loss and polarization loss on EM wave absorption, the dissipation capacity generated by polarization loss and conductivity loss can be calculated separately based on the following formula [47]: (2)$$\varepsilon^{\prime \prime }(\omega )={\varepsilon^{\prime \prime }}_{p}+{\varepsilon^{\prime \prime }}_{c}=({\varepsilon^{\prime \prime }}_{s}-{\varepsilon^{\prime \prime }}_{\infty })\frac{\omega \tau }{1+\omega^{2}\tau^{2}}+\frac{\sigma }{\varepsilon_{0}\omega }$$

The corresponding electromagnetic fitting parameters of the samples are shown in Appendix A. As presented in Figure 4c, the results confirm that the Co@CoP_x_@C-1 composite has the highest conduction loss, while both Co@CoP_x_@C-5 and Co@CoP_x_@C-10 composites have a lower conduction loss than the Co@C composite. Additionally, the polarization loss capability analysis calculated for the Co@CoP_x_@C-1 and Co@CoP_x_@C-5 composites showed a stronger polarization loss capability (Figure 4d). The Tafel curve in Figure 4e shows that the overpotential of the Co@CoP_x_@C-1 composite is much smaller than that of other composites, which means a superior interfacial electron migration ability and the strongest interfacial polarization loss ability. Correspondingly, the interfacial polarization loss capacity relationship of Co@C, Co@CoP_x_@C-5, and Co@CoP_x_@C-10 composites is as follows: Co@CoP_x_@C-5 > Co@C > Co@CoP_x_@C-10, and there is little difference in the interface polarization loss capacity. The Electrochemical Impedance Spectroscopy (EIS) plot in Figure 4f further demonstrates that the impedance of the phosphorylated composite increases, which is not favorable for electron transfer. In a comprehensive analysis, the interfacial polarization loss and conduction loss are not the main polarization loss type of the Co@CoP_x_@C-5 composite. Based on the enhancement of the P-0 dipole and the presence of vacancies, the main polarization loss mechanism originates from the dipole-evoked dipole polarization loss and vacancy-evoked defect polarization loss.

Given that the cobalt metal content will certainly decrease after phosphorylation, which will inevitably make for a decrease in the saturation magnetization of the composites, the magnetic properties of all samples were tested using VSM. As exhibited in Figure 5a, it is evident that the Co@C, Co@CoP_x_@C-1, Co@CoP_x_@C-5, and Co@CoP_x_@C-10 composites have saturated magnetization (Ms) values of 105.5, 58.8, 46.1, and 42.1 emu g^−1^, respectively, inferring that the magnetic storage capacity gradually decreases. For all samples, magnetic loss plays an extremely feeble role in the EM wave absorption process. Modulated by the small real part (μ′) and imaginary part (μ″) values, the magnetic loss is weak to the dissipation of the incident EM waves (Appendix A). In view of the association between the tanδ_ε_ and the tanδ_μ_, a smaller difference will promote the improvement in impedance matching. Analysis of the data in Appendix A yields a significant decrease in the tanδ_μ_ values for the Co@CoP_x_@C-5 and Co@CoP_x_@C-10 composites, whereas the change in the tanδ_μ_ values is insignificant, which indicates that the dielectric loss not only constitutes the main mechanism for the dissipation of the EM waves but also promotes the improvement of the impedance matching.

The magnetic loss can be further distinguished from the magnetic loss mechanism by using C_0_, which is calculated as follows [48,49]:
C_0_ = µ″(µ′)^−2^*f*^−1^(3)

The C_0_ curve in Figure 5b can be used to analyze the specific loss type of magnetic loss. The C_0_ curve of all composites fluctuates in the low-frequency range, indicating that natural resonance is dominant in the low-frequency range. In the middle- and high-frequency range, the C_0_ curve remains almost unchanged, demonstrating the dominance of eddy current loss [50]. On account of the research results on the internal mechanisms, the collaborative efficiency of various loss mechanisms in the absorbents was further analyzed, consisting of the attenuation constant (α) representing the attenuation capacity and the impedance matching standing for the reflection capacity of EM waves on the surface of the absorbents [51]. An explanation of the curve in Figure 5c is that higher attenuation constants of the Co@CoP_x_@C-1 and Co@CoP_x_@C-5 composites demonstrate the synergistic enhancement effect of various loss mechanisms, such as interfacial polarization loss, defect polarization loss, dipole loss, and conduction loss.

Nevertheless, considering that improper impedance matching can cause the reflection of EM waves on the surface of the material, superior attenuation ability does not necessarily precipitate excellent EM wave absorption performance. Commonly, the improvement in impedance matching allows EM waves to enter the material as much as possible, rather than reflecting on the material surface, thereby promoting more EM waves to be attenuated. The associated impedance matching plot (Figure 5d) represents four different |Z_in_/Z_0_| curves. A critical analysis shows that the Z_in_/Z_0_ values of the Co@CoP_x_@C-1 composite are below 1 throughout the entire frequency range, implying that the EM waves are mainly reflected [52]. Furthermore, by analyzing the Z_in_/Z_0_ values for the Co@C, Co@CoP_x_@C-5, and Co@CoP_x_@C-10 composites, it can be concluded that the absorption frequency of the phosphorylated samples significantly shifts the to the left when Z_in_/Z_0_ is equal to 1, which helps to achieve EM wave absorption in low-frequency region.

Theoretically, the RL values of the absorbents depend on the permittivity and permeability; the calculated formula is as follows [53,54]:(4)$$Z_{in}=Z_{0}\sqrt{\frac{\mu_{r}}{\varepsilon_{r}}}\tan h\left[j\frac{2\pi }{c}\sqrt{\mu_{r}\varepsilon_{r}}fd\right]$$
(5)$$R_{L}\left(dB\right)=20\log\left|\frac{Z_{in}-Z_{0}}{Z_{in}+Z_{0}}\right|$$
where ε_r_ and μ_r_ represent complex permittivity and complex permeability, Z represents impedance matching, and RL represents reflection loss. To more visually characterize the EM wave absorption properties of the samples, the 2D plot, λ1/4 curve, 3D plot, EAB curve, and 2D plane show that the prepared Co@CoP@C-5 composites accomplish effective attenuation of EM waves with a minimum reflection loss (RL_min_) of −55 dB and an effective absorption bandwidth (EAB) of 5.5 GHz at a thickness of 2.0 mm, and RL curves also indulge the λ1/4 curve. Data for the comparison samples calculated in Appendix A manifest that the RL_min_ of Co@C, Co@CoP_x_@C-1, and Co@CoP_x_@C-10 are −29.86 dB, −14.66 dB, and −45.38 dB at the thicknesses of 2.5 mm, 2.0 mm, and 3.0 mm, respectively, and the EAB values are up to 6.16, 5.36, and 4.56 GHz, respectively. It can be straightforwardly inferred that the Co@CoP_x_@C-5 composite not only materializes extremely excellent EM wave absorption performance at a thin thickness but also allows for EM wave attenuation over a wide range of frequencies.

The controllable synthesis of anion-rich-vacancy Co@CoP_x_@C composites due to different contents of phosphide reveals the dual-polarization loss mechanisms of action with defective polarization loss and dipole polarization loss dominating the polarization loss capability. The detailed EM wave loss mechanisms are shown in Figure 6. Firstly, the one-dimensional porous nanorod structure has a large specific surface area and porosity, which provides more electron transport channels and facilitates rapid electron hopping and migration, thus increasing the conduction loss. Secondly, three substances with different electron hopping behaviors form an abundant heterogeneous interface containing Co/C, CoP_x_/C, and Co/CoP_x_, which is conducive to achieving enhanced interfacial electron transfer to promote interfacial polarization loss. Thirdly, the phosphorus vacancy defects in CoP_x_ nanorods lead to enhanced defect polarization, and the presence of P-O bonds also promotes the enhancement of defect polarization. In addition, the dipoles in carbon, including C-O/C=O, are the main factors that promote the loss of dipole polarization. Fifthly, the magnetic properties can also further contribute to magnetic loss.

## 4. Conclusions

The controllable regulation of polarization loss in dielectric loss is achieved based on suitable EM parameters. It is demonstrated, relying on extensive reliable characterization, that the optimization of EM parameters to promote a dual-polarization loss strategy is conducive to the enhancement of EM wave absorption performance. Of course, the main influencing factor can be attributed to the fact that CoP_x_ with defects on the uniformly loaded porous carbon structure modulates the interfacial and defective polarization, resulting in a controllable regulation of polarization loss. As a result, the RL_min_ of the Co@CoP_x_@C composites reaches −55 dB at only 2.0 mm, and the EAB is 5.5 GHz. Herein, with an eye to the amorphous CoP_x_ with phosphorus vacancies, we reveal that the defect and interface polarization strategy is beneficial for the regulation of the dielectric constant, thus expanding the research ideas for the application of EM parameter modulation in the field of EM wave absorption.

## Figures and Tables

**Figure 1 nanomaterials-13-03025-f001:**
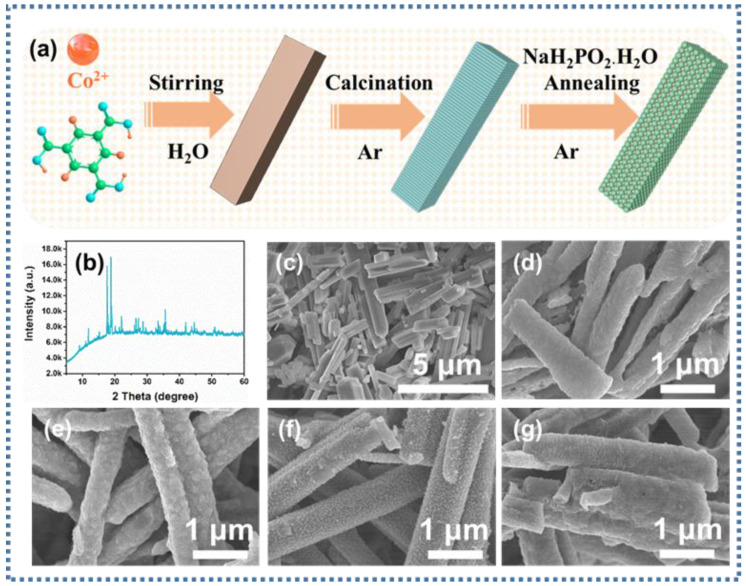
(**a**) Schematic of the synthesis of Co@CoP_x_@C composites; (**b**) XRD spectra of Co-MOF nanorods; SEM images of (**c**) Co-MOF, (**d**) Co@C, (**e**) Co@CoP_x_@C-1, (**f**) Co@CoP_x_@C-5, and (**g**) Co@CoP_x_@C-10 nanorods.

**Figure 2 nanomaterials-13-03025-f002:**
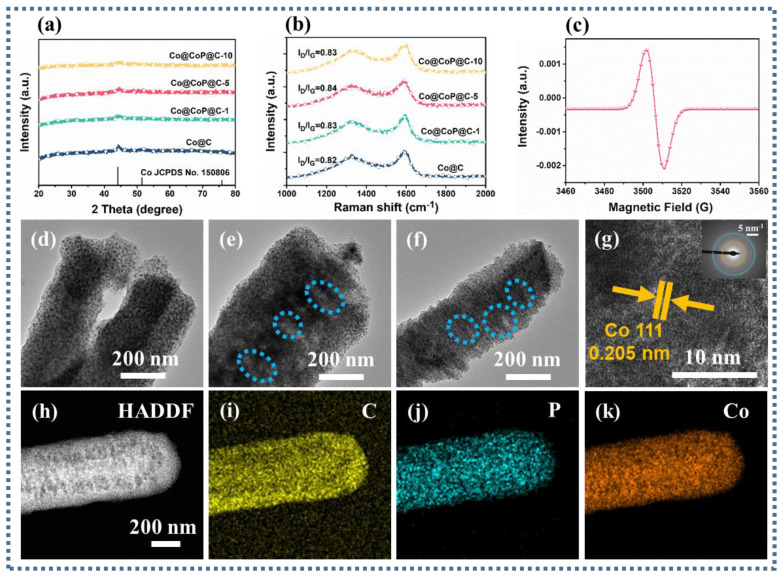
(**a**) XRD spectra, (**b**) Raman spectra, (**c**) EPR, (**d**–**f**) TEM images of the Co@CoP_x_@C-1, Co@CoP_x_@C-5, and Co@CoP_x_@C-10 composites; (**g**) HRTEM image of the Co@CoP_x_@C-10 composites, (**h**) HAADF, and (**i**–**k**) the EDS mapping images of the Co@CoP_x_@C-10 composites.

**Figure 3 nanomaterials-13-03025-f003:**
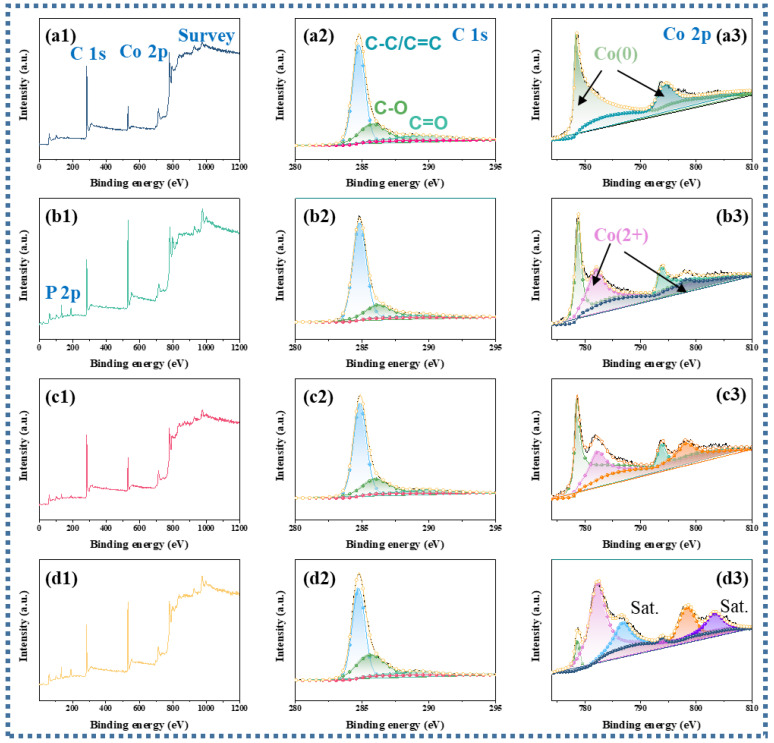
(**a1**–**d1**) XPS survey spectrum, (**a2**–**d2**) high-resolution C 1s, (**a3**–**d3**) high-resolution Co 2p for Co@C, Co@CoP_x_@C-1, Co@CoP_x_@C-5, and Co@CoP_x_@C-10 composites.

**Figure 4 nanomaterials-13-03025-f004:**
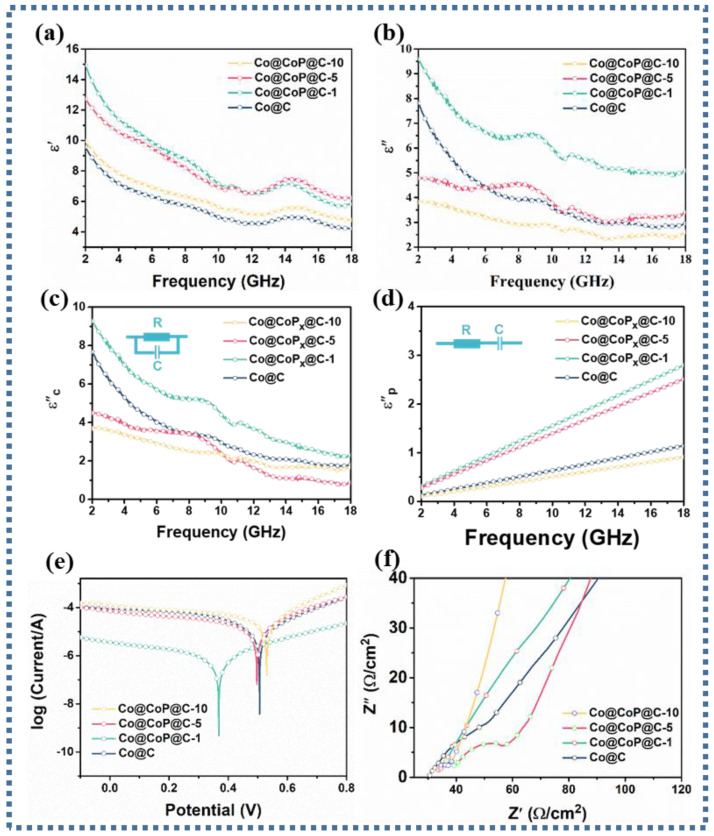
(**a**) Real part ε′, (**b**) imaginary part ε″, (**c**) conduction loss and (**d**) polarization loss, (**e**) Tafel curves, (**f**) Nyquist plots of Co@C, Co@CoP_x_@C-1, Co@CoP_x_@C-5, and Co@CoP_x_@C-10 composites.

**Figure 5 nanomaterials-13-03025-f005:**
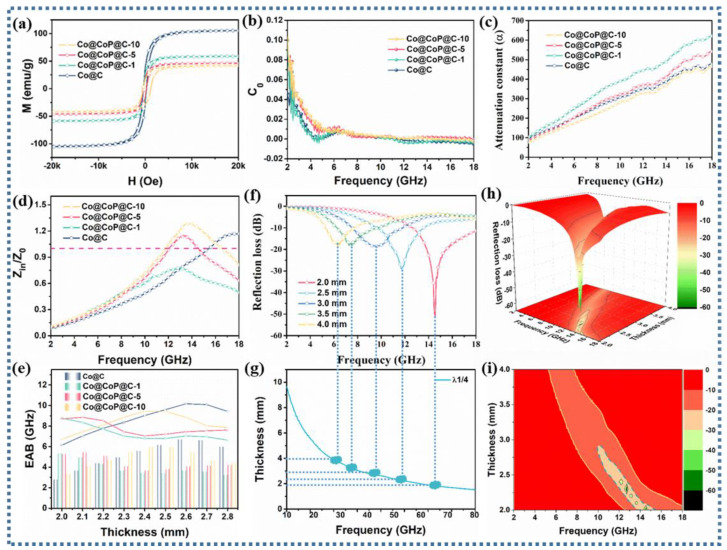
(**a**) Hysteresis loops, (**b**) C_0_ curves, (**c**) attenuation constant (α), (**d**) impedance matching |Z_in_/Z_0_| curves, (**e**) EAB comparison plots, (**f**) 2D RL curves, (**g**) λ1/4 curves, (**h**) and 3D plots and (**i**) 2D surface plots of Co@C, Co@CoP_x_@C-1, Co@CoP_x_@C-5, and Co@CoP_x_@C-10 composites.

**Figure 6 nanomaterials-13-03025-f006:**
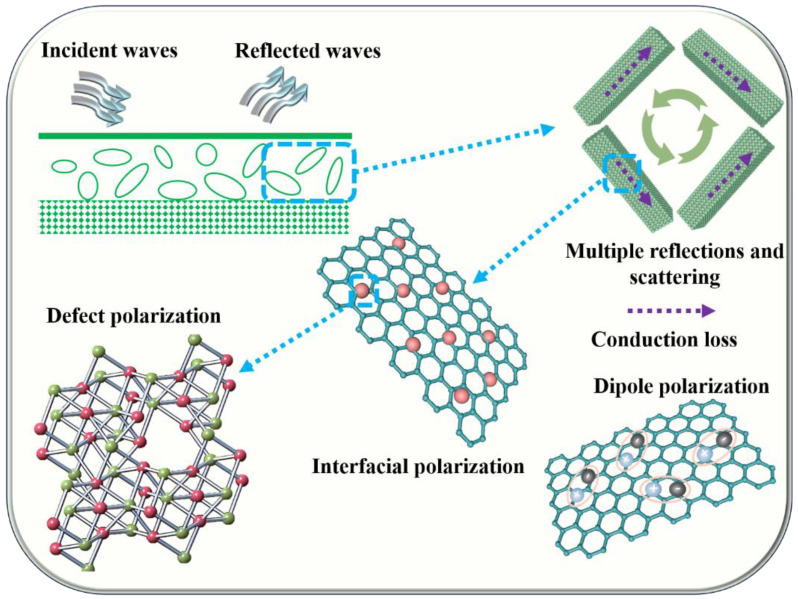
Schematic illustration of the mechanisms of EM wave absorption for amorphous Co@CoP_x_@C composites.

## Data Availability

The data are not publicly available due to privacy or ethical restrictions.

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
