# Peer review of "Tuning Electromagnetic Parameters Induced by Synergistic Dual-Polarization Enhancement Mechanisms with Amorphous Cobalt Phosphide with Phosphorus Vacancies for Excellent Electromagnetic Wave Dissipation Performance"

_nanomaterials, 2023, doi:10.3390/nano13233025_

Round 1

Reviewer 1 Report

Comments and Suggestions for Authors

The paper is devoted for Co@CoPx  @C composites synthesis and characterization. The topic is generally interesting, however the paper contain unexplained places (below) and need major revisions.

Fig. S5 is not correct. The scale for Cole-Cole plot should be the same for the real and imaginary parts of dielectric permittivity (x and y axes).

The equation (1) is not valid for parameters presented in Fig. 4 c and d, the sum of ε’’p and ε’’c is not equal ε’’. In Table S1 εs is much higher than ε’ presented in Fig. 4a, it suggest that 1/τ is out of your frequency range. Therefore, the fit and corresponding discussion is very doubtful.  Why you not presented parameter τ values in Table S1?

The text related to the Fig. 6 should be rewritten. The difference between defect polarization and dipole polarization should be explained in details. The physical meaning of ‘’dipole in carbon’’ should be also explained. Why magnetic losses are ‘’retained’’?

Conclusions should be rewritten in more informative way.

Some relevant references should be cited in the paper text [1-3].

All misprints should be corrected, for example in abstract ‘’realized, Thus promoting ’’.

1.                   I. Vanskevice et al, Materials 15, 876 (2022).

2.                   M. Bleija et al, Nanomaterials 12, 3671 (2022).

3.                   M. Bleija et al, Polymers 15, 515 (2023).

Author Response

Reviewer1

  1. Fig. S5 is not correct. The scale for Cole-Cole plot should be the same for the real and imaginary parts of dielectric permittivity (x and y axes).

Our response: we thank the reviewer for the suggestion. We have modified the Cole-Cole plots.

  1. 2. The equation (1) is not valid for parameters presented in Fig. 4 c and d, the sum of ε’’p and ε’’c is not equal ε’’. In Table S1 εs is much higher than ε’ presented in Fig. 4a, it suggest that 1/τ is out of your frequency range. Therefore, the fit and corresponding discussion is very doubtful. Why you not presented parameter τ values in Table S1?

Our response: we thank the reviewer for the suggestion. To solve the above problems,  We recalculated the polarization loss and conduction loss, and the results are shown in the figure below. Based on the literature (Chem. Eng. J. 2022, 433: 134484), it can be seen that εs is the static dielectric constant, and it is normal that εs is greater than ε′. Besides, we list the value of τ in the
table.

  1. The text related to the Fig. 6 should be rewritten. The difference between defect polarization and dipole polarization should be explained in details. The physical meaning of ‘’dipole in carbon’’ should be also explained. Why magnetic losses are ‘’retained’’?

Our response: we thank the reviewer for the suggestion. We've rewritten the description.

Controllable synthesis of anion-rich vacancy Co@CoPx@C composites due to different content phosphide strategies reveals the dual-polarization loss mechanisms of action with defective polarization loss and dipole polarization loss dominating the polarization loss capability. The detailed EM wave loss mechanisms are shown in Figure 6. Firstly, the one-dimensional porous nanorod structure has a large specific surface area and porosity, which provides more electron transport channels and facilitates rapid electron hopping and migration, thus increasing the conduction loss. Secondly, three substances with different electron hopping behaviors form an abundant heterogeneous interface containing Co/C, CoPx/C and Co/CoPx, which is conducive to achieving enhanced interfacial electron transfer to promote interfacial polarization loss. Thirdly, the phosphorus vacancy defects in CoPx nanorods lead to enhanced defect polarization, and the presence of P-O bonds also promotes the enhancement of defect polarization. In addition, the dipoles in carbon including C-O/C=O are the main factors that promoting the loss of dipole polarization. Fifthly, The magnetic properties can also further contribute to magnetic loss.

Because the phosphating is not complete, the prepared material still has magnetic properties, which can be obtained by testing the saturation magnetization (Figure 5a).

  1. Conclusions should be rewritten in more informative way.

Our response: we thank the reviewer for the suggestion.

The controllable regulation of polarization loss in dielectric loss is achieved based on suitable EM parameters. It is demonstrated, relying on extensive reliable characterization, that the optimization of EM parameters to promote a dual-polarization loss strategy is conducive to the enhancement of EM wave absorption performance. Of course, the main influencing factor can be attributed to the fact that CoPx with defects on the uniformly loaded porous carbon structure modulates the interfacial and defective polarization, resulting in a controllable regulation of polarization loss. As a result, The RLmin of Co@CoPx@C composites reaches -55 dB at only 2.0 mm and the EAB is 5.5 GHz. Herein, with an eye to the amorphous CoPx with phosphorus vacancies, revealing that the defect and interface polarization strategy is beneficial for the regulation of dielectric constant, thus expanding the research ideas for the application of EM parameter modulation in the field of EM wave absorption.

  1. Some relevant references should be cited in the paper text [1-3].

Our response: we thank the reviewer for the suggestion. Some relevant references have been cited in the paper text [1-3].

[1] Vanskevice, I.; Kazakova, M.A.; Macutkevic, J.; Semikolenova, N.V.; Banys, J. Dielectric Properties of Hybrid Polyethylene Composites Containing Cobalt Nanoparticles and Carbon Nanotubes. Materials 2022, 15 (5): 1876.

[2] Bleija, M.; Platnieks, O.; Macutkevic, J.; Starkova, O.; Gaidukovs, S. Comparison of Carbon-Nanoparticle-Filled Poly(Butylene Succinate-co-Adipate) Nanocomposites for Electromagnetic Applications. Nanomaterials 2022, 12 (30): 3671.

[3] Bleija, M.; Platnieks, O.; Macutkevic, J.; Banys, J.; Starkova, O.; Grase, L.; Gaidukovs, S. Poly(Butylene Succinate) Hybrid Multi-Walled Carbon Nanotube/Iron Oxide Nanocomposites: Electromagnetic Shielding and Thermal Properties. Polymers 2023, 15 (3): 515.

  1. All misprints should be corrected, for example in abstract ‘’realized, Thus promoting ’’.

Our response: we thank the reviewer for the suggestion. All the misprints have been corrected.

  1. is successfully synthesized, thus adjusting.
  2. multi-component and complex structures.
  3. and conductive polymers.
  4. contribute to favorable conductive loss.
  5. is conducive to further increasing.
  6. with rougher surfaces.
  7. Co@ CoPx @C composites.
  8. were analyzed using.
  9. is authoritative in.
  10. inferring that the magnetic storage.
  11. As a result, the RLmin…”

Reviewer 2 Report

Comments and Suggestions for Authors

SUMMARY

In the paper, the authors investigate the development and characterization of Co@CoP x @C composites designed for electromagnetic wave (EMW) absorption. It starts by outlining the importance of electromagnetic (EM) wave absorption in preventing interference and potential health hazards caused by electromagnetic waves. The study delves into various mechanisms contributing to EM wave attenuation, encompassing conduction, magnetic, and dielectric losses, along with multiple reflection and scattering phenomena. The study chronicles a literature review that underscores recent advancements in this domain, focusing on the evolution of materials and revealing avant-garde EM wave attenuation mechanisms. It includes insights into the properties and functions of diverse materials, from conductive polymers to magnetic elements and alloys, with a strong emphasis on enhancing EM wave absorption by regulating multi-component and complex structures.
The experimental results and subsequent discussion illuminate the step-by-step synthesis of Co@CoPx@C composites, with thorough characterizations employing a spectrum of techniques. The analyses include XRD, SEM, TEM, Raman spectroscopy, and EPR, elucidating the structural changes, surface modifications, and alterations in dielectric and magnetic properties brought by phosphating the composites.
The study rigorously evaluates the EM wave absorption performance of the developed composites. It includes a detailed examination of reflection loss (RL), effective absorption bandwidth (EAB), and impedance matching analyses. The article concludes by highlighting the achievements in controllable regulation of polarization loss in dielectric materials, leading to the optimization of EM parameters for enhanced EM wave absorption performance. The development of Co@CoPx@C composites demonstrates a minimum reflection loss (RL min) of –55 dB at a thickness of 2.0 mm and an effective absorption bandwidth (EAB) of 5.5 GHz. The unique characteristics of amorphous CoPx with phosphorus vacancies are noted as pivotal for further advances in regulating EM parameters and improving impedance matching.

POSITIVE ASPECTS

1. Based on a literature review, the authors reviewed various aspects related to electromagnetic (EM) wave absorption in the introduction section.
2. The authors noted the less explored but promising use of transition metal phosphide heterojunctions, emphasizing their excellent properties attributed to charge transfer at interfaces, complex heterogeneous structures, and abundant edge activity.
3. The authors presented their own work involving the synthesis of amorphous Co@CoP x @C composites with phosphorus vacancies. They highlighted the role of these vacancies in enhancing interfacial polarization loss, promoting defect polarization, and the importance of phosphorus-doped porous carbon in achieving better polarization loss.
4. The authors conducted a comprehensive analysis and discussion of the experimental outcomes related to the synthesis and characterization of the Co@CoP x @C composites, focusing on various aspects.
5. The authors outlined a multi-step process for synthesizing the Co@CoP x @C composites, demonstrating the formation of the materials from Co-MOF nanorods through carbonization and subsequent phosphating. They presented schematics, XRD spectra, and SEM images to depict the structural evolution through each synthesis step.
6. The authors evaluated the EM wave absorption properties using reflection loss (RL), effective absorption bandwidth (EAB), and impedance matching analyses.
7. Through schematics and discussion, the authors elucidated the mechanisms responsible for EM wave absorption in the Co@CoP x @C composites. They highlighted multiple factors contributing to EM wave attenuation, such as conductivity loss, interfacial polarization, defect polarization, dipole polarization, and magnetic loss.

CONCERNS

The presented work is useful but has some issues that need to be removed. Points that must be addressed by authors are listed below:

Major concerns
1. Authors should write out the meaning of the acronyms EMW, MOF, PDF, and EIS upon first usage. Correct accordingly throughout the article.
2. The manuscript does not contain the Materials and Methods section. Complete the Materials and Methods section in the manuscript.
3. The manuscript does not contain an experimental section, although the authors refer to this section in the text.
4. In the text, the authors refer to Fig. 1, which does not exist in the manuscript. Compare the description “Scheme 1” versus, for example, “Fig. 1(b)”.
5. The authors refer to Figure S1 in the text, but it is not in the manuscript. Similarly, it applies to Figure S2, Figure S3, Figure S4, Figure S5. Add the pictures in the supplementary file to the manuscript.

Minor concerns
1. The authors use a capital letter after a comma in some sentences, for example: “…with phosphorus vacancy is successfully realized, Thus promoting…” or “As a result, The RLmin…” Check-in detail similar typos in the entire manuscript.
2. The physical quantity symbols are always written in italic (sloping) type, irrespective of the type used in the rest of the text (ISO 80000-1: 2009), ISO 80000-5: 2007).

REMARKS

1. The authors used the word “Second” twice in a row, which is confusing when reading the last paragraph in the Results and Discussion section.
2. In the Conclusion section, I lack a vision for further research.

RECOMMENDATIONS

1. I think it is necessary to correct the error in the following sentence: “The relationship between ε′ and ε′ can be described by the…”  Similarly, correct the error in the next sentence.

QUESTIONS

I have two questions for the authors of the article.
1. What is the difference between reflection loss -55 dB and --55 dB?
2. What is the environmental impact of Co@CoPx@C and the overall production technology of this composite? What is the effect of Co@CoPx@C on living organisms? Add to the manuscript information about the environmental burden during the production of Co@CoPx@C and the health impact on living organisms.

Answer the given questions with comments in the manuscript.

CONCLUSION

I find this article helpful. Regretfully, the paper cannot be accepted in its present form. The authors of the present article have to correct the issues.

Author Response

Reviewer2

Major concerns

  1. Authors should write out the meaning of the acronyms EMW, MOF, PDF, and EIS upon first usage. Correct accordingly throughout the article.

Our response: we thank the reviewer for the suggestion. We have corrected accordingly throughout the article.

  1. The manuscript does not contain the Materials and Methods section. Complete the Materials and Methods section in the manuscript.

Our response: we thank the reviewer for the suggestion. We have added the Materials and Methods section to the manuscript.

  1. The manuscript does not contain an experimental section, although the authors refer to this section in the text.

Our response: we thank the reviewer for the suggestion. We have added the experimental section to the manuscript.

  1. In the text, the authors refer to Fig. 1, which does not exist in the manuscript. Compare the description “Scheme 1” versus, for example, “Fig. 1(b)”.

Our response: we thank the reviewer for the suggestion. We have modified Scheme 1 into Figure 1.

Figure 1. (a) Schematic of the synthesis of Co@CoPx@C composites; (b) XRD spectra of Co-MOF nanorods; SEM images of (c) Co-MOF, (d) Co@C, (e) Co@CoPx@C-1, (f) Co@CoPx@C-5 and (g) Co@CoPx@C-10 nanorods.

  1. The authors refer to Figure S1 in the text, but it is not in the manuscript. Similarly, it applies to Figure S2, Figure S3, Figure S4, Figure S5. Add the pictures in the supplementary file to the manuscript.

Our response: we thank the reviewer for the suggestion. Unimportant Figures can be placed in the supporting material to prove their general information, so these diagrams remain in the supporting information.

Minor concerns

  1. The authors use a capital letter after a comma in some sentences, for example: “…with phosphorus vacancy is successfully realized, Thus promoting…” or “As a result, The RLmin…” Check-in detail similar typos in the entire manuscript.

Our response: we thank the reviewer for the suggestion. We have revised these typos in the entire manuscript.

  1. is successfully synthesized, thus adjusting.
  2. multi-component and complex structures.
  3. and conductive polymers.
  4. contribute to favorable conductive loss.
  5. is conducive to further increasing.
  6. with rougher surfaces.
  7. Co@ CoPx @C composites.
  8. were analyzed using.
  9. is authoritative in.
  10. inferring that the magnetic storage.
  11. As a result, the RLmin…”
  12. The physical quantity symbols are always written in italic (sloping) type, irrespective of the type used in the rest of the text (ISO 80000-1: 2009), ISO 80000-5: 2007).

Our response: we thank the reviewer for the suggestion. We have completed the revisions in the manuscript.

REMARKS

  1. The authors used the word “Second” twice in a row, which is confusing when reading the last paragraph in the Results and Discussion section.

Our response: we thank the reviewer for the suggestion. We have revised these errors.

  1. In the Conclusion section, I lack a vision for further research.

Our response: we thank the reviewer for the suggestion. We have revised our conclusions.

The controllable regulation of polarization loss in dielectric loss is achieved based on suitable EM parameters. It is demonstrated, relying on extensive reliable characterization, that the optimization of EM parameters to promote a dual-polarization loss strategy is conducive to the enhancement of EM wave absorption performance. Of course, the main influencing factor can be attributed to the fact that CoPx with defects on the uniformly loaded porous carbon structure modulates the interfacial and defective polarization, resulting in a controllable regulation of polarization loss. As a result, The RLmin of Co@CoPx@C composites reaches -55 dB at only 2.0 mm and the EAB is 5.5 GHz. Herein, with an eye to the amorphous CoPx with phosphorus vacancies, revealing that the defect and interface polarization strategy is beneficial for the regulation of dielectric constant, thus expanding the research ideas for the application of EM parameter modulation in the field of EM wave absorption.

RECOMMENDATIONS

  1. I think it is necessary to correct the error in the following sentence: “The relationship between ε′ and ε′ can be described by the…” Similarly, correct the error in the next sentence.

Our response: we thank the reviewer for the suggestion. We have rephrased this sentence. The ε′-ε" can be expressed as the following equation [44-46].

QUESTIONS

I have two questions for the authors of the article.

  1. What is the difference between reflection loss -55 dB and --55 dB?

Our response: we thank the reviewer for the suggestion. This is a typo. It has been corrected.

  1. What is the environmental impact of Co@CoPx@C and the overall production technology of this composite? What is the effect of Co@CoPx@C on living organisms? Add to the manuscript information about the environmental burden during the production of Co@CoPx@C and the health impact on living organisms.

Our response: we thank the reviewer for the suggestion.

In this paper, a milder way of phosphating the product is used that does not produce large amounts of wastewater and solid waste. In addition, due to partial phosphatization, it also reduces the emission of waste gas.

The purpose of the Co@CoPx@C composites is to reduce the hazards of electromagnetic waves, after partial phosphating, the impact of cobalt metal radioactivity on living organisms and the impact on crop growth can be reduced.

CONCLUSION

I find this article helpful. Regretfully, the paper cannot be accepted in its present form. The authors of the present article have to correct the issues.

Round 2

Reviewer 1 Report

Comments and Suggestions for Authors

Authors make proper corrections according to reviewer remarks and I suggest to publish the paper as it is.

Reviewer 2 Report

Comments and Suggestions for Authors

SUMMARY

In the paper, the authors investigate the development and characterization of Co@CoP x @C composites designed for electromagnetic wave (EMW) absorption. It starts by outlining the importance of electromagnetic (EM) wave absorption in preventing interference and potential health hazards caused by electromagnetic waves. The study delves into various mechanisms contributing to EM wave attenuation, encompassing conduction, magnetic, and dielectric losses, along with multiple reflection and scattering phenomena. The study chronicles a literature review that underscores recent advancements in this domain, focusing on the evolution of materials and revealing avant-garde EM wave attenuation mechanisms. It includes insights into the properties and functions of diverse materials, from conductive polymers to magnetic elements and alloys, with a strong emphasis on enhancing EM wave absorption by regulating multi-component and complex structures.
The experimental results and subsequent discussion illuminate the step-by-step synthesis of Co@CoPx@C composites, with thorough characterizations employing a spectrum of techniques. The analyses include XRD, SEM, TEM, Raman spectroscopy, and EPR, elucidating the structural changes, surface modifications, and alterations in dielectric and magnetic properties brought by phosphating the composites.
The study rigorously evaluates the EM wave absorption performance of the developed composites. It includes a detailed examination of reflection loss (RL), effective absorption bandwidth (EAB), and impedance matching analyses. The article concludes by highlighting the achievements in controllable regulation of polarization loss in dielectric materials, leading to the optimization of EM parameters for enhanced EM wave absorption performance. The development of Co@CoPx@C composites demonstrates a minimum reflection loss (RL min) of –55 dB at a thickness of 2.0 mm and an effective absorption bandwidth (EAB) of 5.5 GHz. The unique characteristics of amorphous CoPx with phosphorus vacancies are noted as pivotal for further advances in regulating EM parameters and improving impedance matching.

POSITIVE ASPECTS

1. Based on a literature review, the authors reviewed various aspects related to electromagnetic (EM) wave absorption in the introduction section.
2. The authors noted the less explored but promising use of transition metal phosphide heterojunctions, emphasizing their excellent properties attributed to charge transfer at interfaces, complex heterogeneous structures, and abundant edge activity.
3. The authors presented their own work involving the synthesis of amorphous Co@CoP x @C composites with phosphorus vacancies. They highlighted the role of these vacancies in enhancing interfacial polarization loss, promoting defect polarization, and the importance of phosphorus-doped porous carbon in achieving better polarization loss.
4. The authors conducted a comprehensive analysis and discussion of the experimental outcomes related to the synthesis and characterization of the Co@CoP x @C composites, focusing on various aspects.
5. The authors outlined a multi-step process for synthesizing the Co@CoP x @C composites, demonstrating the formation of the materials from Co-MOF nanorods through carbonization and subsequent phosphating. They presented schematics, XRD spectra, and SEM images to depict the structural evolution through each synthesis step.
6. The authors evaluated the EM wave absorption properties using reflection loss (RL), effective absorption bandwidth (EAB), and impedance matching analyses.
7. Through schematics and discussion, the authors elucidated the mechanisms responsible for EM wave absorption in the Co@CoP x @C composites. They highlighted multiple factors contributing to EM wave attenuation, such as conductivity loss, interfacial polarization, defect polarization, dipole polarization, and magnetic loss.

CONCLUSION

The authors revised the manuscript in accordance with all the recommendations made, therefore I recommend the publication in Nanomaterials.